# Influence of Substituents in a Six-Membered Chelate Ring of HG-Type Complexes Containing an N→Ru Bond on Their Stability and Catalytic Activity

**DOI:** 10.3390/molecules28031188

**Published:** 2023-01-25

**Authors:** Kirill A. Vasilyev, Alexandra S. Antonova, Nikita S. Volchkov, Nikita A. Logvinenko, Eugeniya V. Nikitina, Mikhail S. Grigoriev, Anton P. Novikov, Vladimir V. Kouznetsov, Kirill B. Polyanskii, Fedor I. Zubkov

**Affiliations:** 1Organic Chemistry Department, Faculty of Science, Peoples’ Friendship University of Russia (RUDN University), Miklukho-Maklaya St., 6, Moscow 117198, Russia; 2Frumkin Institute of Physical Chemistry and Electrochemistry, Russian Academy of Sciences, Leninsky pr. 31, bld. 4, Moscow 119071, Russia; 3Laboratorio de Química Orgánica y Biomolecular, Escuela de Química, Universidad Industrial de Santander, Cl. 9 # Cra 27, Bucaramanga 680006, Colombia

**Keywords:** ruthenium complexes, Hoveyda-Grubbs-type catalysts, ring-closing metathesis (RCM), nitrogen-ruthenium coordinate bond, six-membered chelate cycle, [1,3-bis(2,4,6-trimethylphenyl)imidazolidin-2-ylidene](dichloro)[benzylidene]ruthenium

## Abstract

An efficient approach to the synthesis of olefin metathesis HG-type catalysts containing an N→Ru bond in a six-membered chelate ring was proposed. For the most part, these ruthenium chelates can be prepared easily and in high yields based on the interaction between 2-vinylbenzylamines and **Ind II** (the common precursor for Ru-complex synthesis). It was demonstrated that the increase of the steric volume of substituents attached to the nitrogen atom and in the α-position of the benzylidene fragment leads to a dramatic decrease in the stability of the target ruthenium complexes. The bulkiest *^i^*Pr substituent bonded to the nitrogen atom or to the α-position does not allow the closing of the chelate cycle. N,N-Diethyl-1-(2-vinylphenyl)propan-1-amine is a limiting case; its interaction with **Ind II** makes it possible to isolate the corresponding ruthenium chelate in a low yield (5%). Catalytic activity of the synthesized complexes was tested in RCM reactions and compared with α-unsubstituted catalysts obtained previously. The structural peculiarities of the final ruthenium complexes were thoroughly investigated using XRD and NMR analysis, which allowed making a reliable correlation between the structure of the complexes and their catalytic properties.

## 1. Introduction

In the last decade of the 20th century, the olefin metathesis took a worthy place among the most important reactions of organic chemistry and in industrial processes [1]. To date, along with common Hoveyda–Grubbs (HG) catalysts with a coordination bond O→Ru in a five-membered cycle, their heteroanalogues containing S→Ru, P→Ru, and N→Ru chelate bonds have been described.

The history of nitrogen-containing ruthenium derivatives including a six-membered chelate cycle began in 2000, when Van Der Scaaf proposed a simple synthesis of 2-pyridylethanyl substituted ruthenium carbene complexes [2]. Afterwards, in 2004, the highly-reactive ruthenium complexes suitable for different metathesis reactions were obtained by Schrodi, who was the first to introduce an N-heterocyclic carbene ligand into the structure of six-membered N→Ru chelates [3]. Since then, changes of substituents around the heteroatom [4,5,6,7,8,9,10,11], the fine tuning of NHC ligands [6,12,13,14,15,16], and the introduction of different functional groups into the benzylidene moiety of the chelate ring have become the main directions of improvement of practical and useful properties of aza-HG-type catalysts [8,9,14,15,17,18,19]. For example, such types of ruthenium derivatives were successfully applied in the synthesis of natural compounds [20,21] and for stereoselective CM reactions [22,23,24,25].

Although the HG-catalysts containing an *O*→Ru bond in a five-membered chelate cycle remain among the most popular, both in synthetic practice and in industry [26,27], nitrogen-bearing ruthenium complexes begin to play an increasingly prominent role. For instance, at the beginning of the 21st century, the third-generation Grubbs catalysts (like **G-III-Br**, Figure 1) were implemented in laboratory practice. It should also be mentioned that six-membered N→Ru complexes are successfully used not only in fine synthesis for RCM and ROMP reactions [28,29,30,31,32,33], but they have also been recently patented for commercial use including oil refining, gasoline reforming, and as catalysts for metathesis polymerization of dicyclopentadiene (DCPD) [34,35,36,37,38].

Despite the fact that the six-membered N-ruthenium chelates are less popular in comparison with their five-membered homologues, they have some important advantages. In particular, they possess an additional site for structural modification due to the presence of an α-position relative to the nitrogen atom in a benzylidene moiety [28,29,34,35,36] (see the bottom row in Figure 1). In the previous communications [28,29], it had been proved that catalytic properties of these N→Ru chelates are closely dependent on the steric volume of substituents surrounding the donating nitrogen atom (in Figure 1, the three most active catalytic complexes are shown). As a rule, with an increase in the volume of N-substituents, the catalytic activity increased, but the stability of the complexes fell. Thereby, this work is the logical continuation of our previous articles [28,29]; in particular, here we investigate the influence of the volume of the substituent in the α-position (Figure 1) on the possibility of the closing of the N→Ru chelate ring, as well as on the catalytic properties of the resulting N→Ru complexes. In the beginning of this work, it was planned to introduce methyl, ethyl, isopropyl, and *tert*-butyl substituents into the α-position, simultaneously varying the volume of the radicals at the nitrogen atom.

## 2. Results and Discussion

### 2.1. Synthesis of the Initial 2-vinylbenzylamines 1, 2, and 3

In the beginning, we needed to synthesize the nitrogen-containing ligands for the assembly of the benzylidene unit of the Ru-complexes (Figure 1, Figure 2 and Figure 3). Based on the available starting compounds, two close synthetic routes to the target 2-vinyl benzylamines (**1**–**3**) have been developed. Note that both methods can be used on an industrial scale.

Synthesis of styrenes **1** and **2** was carried out by acylation of inexpensive phenethylamine, followed by the Bischler–Napieralski cyclization of the obtained amides [39]. The resulting 1-methyl and 1-isopropyl-3,4-dihydroisoquinolines were sequentially alkylated with dialkyl sulfates and reduced to give the corresponding 1-methyl and 1-isopropyl-N-methyl-1,2,3,4-tetrahydroisoquinolines (THQs) [29]. Further, the repeated quaternization of the nitrogen atom was carried out, followed by one-pot with the Hofmann cleavage (Figure 1) [28,40]. The total yields of styrenes **1a**–**c** and **2a**–**c** obtained according to this approach were 42–56% based on the initial phenethylamine. Yields of **1** and **2** after the last two stages are given in Table 1 (based on 1-alkyl-3,4-dihydroisoquinolines).

Interestingly, the above-described method proved to be unsuitable for obtaining the N-isopropyl substituted benzylamines, similar to **1e** (Figure 2). Presumably, this is due to a significant steric volume of the *^i^*Pr group and the N-alkylation step passes in a low yield.

Advantages of the styrene synthesis, proposed in Figure 1, lie not only in the availability of the initial and auxiliary reagents but also in the scalability of all stages, which do not include the use of absolute or highly toxic reagents. In other words, this method is optimal from the point of view of the industrial synthesis of HG-catalysts. On the other hand, more suitable reagents are available for laboratory synthesis that enable the reduction in the number of stages.

An alternative synthetic approach has been elaborated on for the preparation of α-ethyl substituted styrenes **3** (Figure 3). In this process, available 3,4-dihydroisoquinoline was quaternized according to a standard procedure [29] to give N-alkyl-3,4-dihydroisoquinoline salts. The second C-alkylation of the C=N double bond by the Grignard reagent leads to the corresponding THQs. The last ones were introduced in two successive one-pot steps, the alkylation and the Hofmann cleavage, to form the target 2-aminomethylstyrenes **3a**–**c** in overall yields greater than 60% (Table 1) [28,40]. The total yields of **3a**–**c** after all steps, according to Figure 2, were 68–75% based on 3,4-dihydroisoquinoline.

The last two styrenes, **1d** and **3d**, possessing a morpholine fragment were obtained using the third synthetic approach presented in Figure 4. After N-protection, α-alkylation, and the deprotection of N-Boc-THQ, the intermediate 1-R^3^-substituted THQs were quaternized with 1-chloro-2-(2-chloroethoxy)ethane. The one-pot Hofmann reaction completed the sequence and provided the target benzylamines **1d** and **3d** in good yields (Table 1) [40,41]. The total reaction yields of **1d** and **3d** based on 1,2,3,4-tetrahydroisoquinoline were 85 and 81%, respectively.

Thus, all approaches to the synthesis of styrenes **1**–**3** shown in Figure 1, Figure 3 and Figure 4 complement each other. Depending on the substituents needed and the available starting reagents, a chemist can choose the most convenient route.

After purification with column chromatography, all 2-vinylbenzylamines **1**–**3** were obtained as yellowish oily liquids.

### 2.2. Synthesis of Ruthenium Complexes

For the synthesis of metallo-complexes, ligands **1**–**3** obtained in the first section were introduced into the reaction with precursor **Ind II** [28,42,43] and were frequently used for preparation of ruthenium chelates (Figure 5). Boiling toluene turned out to be a good solvent for the preparation of α-methyl-substituted **4a**–**c**, but the yields of the other ruthenium derivatives **4**–**6** were unsatisfactory under the same conditions. Therefore, the conditions were optimized with varying solvents, temperature, and duration of synthesis (see the ESI). A mixture of toluene/heptane in the V/V ratio of 1:5 at 110 °C proved to be the best medium for the synthesis of the compounds **4d** and **5a**–**d** (Figure 5). Under both mentioned conditions, a complex was completed in less than 1.5 h (Table 2). Catalysts **4a**–**c** and **5a**–**c**, with α-methyl or α-ethyl group, usually precipitated from solutions after cooling of the reaction mixtures to −20 °C (for detailed information, see the ESI).

It should be noted that we were not able to isolate the ruthenium complexes **6** (Table 2) synthesized on the base of α-isopropyl benzylamines **2a**–**c** (Table 1). Although they probably exist in solutions, which is confirmed with TLC, they appear as bright green spots on a chromatographic plate. These complexes are poorly crystallized after cooling of the reaction mixtures, and the resulting precipitates are soluble even in ice-cold pentane. After numerous attempts, we could not isolate the target isopropyl-substituted chelates **6** in a pure form.

According to previously published data [28,29], the morpholine-containing ruthenium complex **N-Morph** (see Figure 1) exhibits one of the best activities towards metathesis reactions. To improve these useful properties, in this work, we inserted a substituent in the α-position of the morpholine-substituted ruthenium chelate. We assumed that the steric volume of any substituent R^3^ next to the ruthenium atom will loosen the N→Ru donor-acceptor bond, and that should help to increase catalytic properties. Formation of chelates **4d** and **5d** proceeded faster than similar transformations, providing analogues **4a**–**c** and **5a**–**c** (Figure 5, Table 2).

Complexes with methyl and ethyl groups in the α-position were isolated in good yields, except for **5c** (Table 2). Attempts to change the solvent with absolute hexane, heptane, benzene, toluene, or their mixtures did not lead to an increase in the yield of the target product. In the best cases, the yield of compound **5c** did not exceed 5%.

From the data of Table 2, it can be concluded that ruthenium chelates **6** with an isopropyl group in the α-position are unstable and could not be isolated in any case (even N,N-dimethyl substituted **6a** cannot be synthesized). Presumably, the simultaneous presence of three ethyl groups in the benzylidene moiety of the complex **5c** was the limiting case for the closing of the chelate ring. With a high probability, more sterically-loaded coordination compounds of type **4**, **5** could not be assembled. 

All catalysts **4a**–**d**, **5a**–**d** were obtained as emerald green powders. The majority of them crystallized readily from a mixture of heptane/dichloromethane or chloroform/dichloromethane, with the formation of well-shaped crystals. An exception was catalyst **5c**, which, in contrast to the others, was fairly soluble in nonpolar solvents, such as pentane and hexane. In this regard, this substance is quite difficult to isolate in a pure from.

The structure of compounds **4**–**5** were established using NMR and, in the cases of **4a**,**b**,**c**, and **5a**,**b**,**d**, using XRD analysis. The most well-defined signals of the key Ru=CH fragment were observed in the range of δ 18.70–19.10 ppm in ^1^H NMR and 312–324 ppm in ^13^C NMR spectra. Such downfield shifts are usual for chelates with a coordinating N→Ru bond [8,28,29,36] and are explained by the strong polarization of the Ru=C bond towards the metal atom. Otherwise, the obtained NMR spectra do not differ from those already described earlier for homologues [28,29]. It should be noted that the signals of the mesityl fragments (Mes) in both the proton and carbon spectra in some compounds turn out to be strongly broadened due to a slow rotation of the Mes-moieties around the C–N bond. To overcome this difficulty, the spectra of such ruthenium derivatives were additionally recorded in dichloromethane (CD_2_Cl_2_).

### 2.3. XRD Analysis of Ruthenium Complexes

All obtained structures for **4a**–**c** and **5a**,**b** contained one independent molecule of the ruthenium complex, except for structure **5d** where there were two crystallographically independent molecules of the complex with the same conformation. One ethyl group in **4c** is disordered. In all complexes, the chlorine atoms were coordinated to the ruthenium atom in the *trans*-position (Cl atoms are on opposite sides of the N2Ru1C2 plane). The Cl-Ru-Cl angles varied from 157.83° to 168.22° (see Table 3). The Ru–Cl distances are close to those for the other N→Ru HG-type *trans*-catalysts [28,29] described earlier and are not of particular interest for discussion. All complexes **4** and **5** contained intramolecular hydrogen bonds of the C-H···Cl type (see the ESI).

In complexes **4** and **5**, the six-membered rings containing N→Ru coordination bond had a distorted conformation. As in the previously obtained analogs [28,29] in **4**, **5a**, and **5b**, the Ru atom and four carbon atoms formed one RuC_4_ plane (the maximum displacement is 0.08 Å), but the sixth pyramidal nitrogen atom deviated from it and formed another RuNC plane. The angle between the RuC_4_/RuNC planes varied from 55.26° to 58.84° and was close to the previously described analogs [28,29]. In the sing crystal of **5d** (in contrast to the other **4**, **5** and previously described complexes of this type) two other planes, CRuNC and C_4_, were formed (the maximum deviation is 0.05 Å) with an angle between the planes of 46.75° and 47.75° for two molecules.

The Ru–C bond length (2.01–2.06 Å) in all compounds differed insignificantly and were close to what was published earlier [28,29]. From the point of view of potential activity in metathesis reactions, the *N*→Ru bond in complexes **4**, **5a**, and **5b** had practically the same length (2.26–2.27 Å). In ethyl-morpholine substituted **5d**, an elongation of this bond (2.35, 2.37 Å) was observed. This result is close to the previously described one for the N-chelate ruthenium complex with a morpholine moiety (**N-Morph**, Figure 1) but without additional α-methyl or α-ethyl groups [28]. The observed Ru–N bond elongation can affect the catalytic activity of the **5d** complex [28,29] (Table 3). 

It is also worth noting that two chelates, **4b** and **5b**, contain two chiral centers each (the *sp*^3^-hybridized nitrogen atom and the α-carbon atom) and, thus, can exist as a pair of diastereomers (Figure 2). According to ^1^H NMR data of the crude reaction mixtures, only one diastereomer of **4b** and **5b** was formed and observed in all experiments. As it was described earlier (Table 3), its spatial structures were unambiguously established with XRD analysis. In the both structures, the bulkiest vicinal substituents (N-Et/C-Me in **4b** and N-Et/C-Et in **5b**) occupied the most favorable pseudoequatorial positions in the six-membered ruthenium-containing ring. The aforementioned fragment was noticeably flattened and had a slightly distorted envelope conformation. With the exception of the nitrogen, all of its atoms practically laid in one plane (the maximum displacement is 0.08 Å for **5b**). Figure 2 clearly shows that the smallest substituents (N-Me/C-H in **4b** and N-Me/C-H in **5b**) were pseudoaxial and occupied the *anti*-periplanar positions.

From the data of XRD analysis, it can be concluded that the addition of methyl or ethyl groups to the α-position relative to the nitrogen atom did not lead to strong changes in the structure of ruthenium complexes. Only the addition of the morpholine fragment resulted in appreciable changes in the structure.

### 2.4. Evaluation of Catalytic Properties

In the final part of the work, the catalytic properties of the resulting ruthenium complexes **4–5** were studied in the simplest ring-closing metathesis (RCM) reactions to evaluate its efficiency. For convenience, the rate of initiation and transformation of starting alkenes into metathesis products will be hereinafter referred to as the catalytic activity. 

Thus, the most common substrate, N,N-diallyltosylamide (**7**), was chosen for the RCM test experiments (Figure 6). The reaction was carried out according to the previously described procedure [28]. After adding the catalyst sample to a tosylate solution, aliquots were taken at a certain time interval. Then, an excess of a catalytic “poison”, ethyl vinyl ether (10% solution in THF), was immediately added to each aliquot to stop the metathesis reaction and to prevent all possible post-transformations. Next, volatile products were removed and the conversion was analyzed with ^1^H NMR. Based on the NMR data, kinetic curves were plotted (Figure 3, Figure 4, Figure 5 and Figure 6).

Simultaneously, the catalytic efficiency of the new complexes **4**–**5** was compared with a range of commercially available **HG-II** [44] or previously reported **N-Morph**, **NEt_2_** (see Figure 1) catalysts of the same type. To achieve this goal, the two most different chelates, **4a** and **4d**, were chosen to describe the efficiency of the whole series of ruthenium derivatives. The activity of N-heterocyclic carbene complexes **4a** and **4d** in RCM reactions with diene **7** turned out to be excellent and exceeded that of the reference catalysts (Figure 3). It was shown that the RCM reaction proceeded reliably with 0.005 mol% catalyst loadings; however, it was more convenient to use higher catalyst concentrations to plot kinetic curves. The reverse side of the high activity was the elevated sensitivity of the solutions of complexes **4**–**5** to the moisture and air (in the crystalline state, the resulting chelates are stable when stored in air). In this regard, all metathesis reactions hereinafter were carried out under an argon atmosphere using dry solvents.

Furthermore, the comparison of the activity of all N→Ru chelates **4**–**5** synthesized in this work was carried out. However, a difference in efficiency between the complexes turned out to be difficult to detect under the aforementioned conditions (0.1 mol% of a catalyst). The data in Figure 4 permit the claim that only the morpholine -containing compounds **4d** and **5d** have a higher initiation rate compared with the other ones, due to an increased steric environment of the donating nitrogen atom (Figure 4).

To overcome the difficulty noted above, the catalyst amount was reduced to 0.01 mol%. In this case, as can be clearly seen in Figure 5, all catalysts **4**–**5** were divided into three groups, in accordance with the size of the substituents around the catalytic center. Similarly, as with the previous results (see Figure 4), complexes **4d** and **5d** possessing bulky morpholine moieties manifested the highest efficiency. The second group consists of α-ethyl-substituted complexes **5a**, **5b**, and **5c**, and the worst activity is exhibited by α-methyl-containing catalysts **4a**, **4b**, and **4c**. As expected in the beginning of this work, an increase in the steric volume of the substituent in the α-position led to an increase in catalytic activity, albeit insignificantly. However, the difference between the complexes **4a**, **4b**, **4c**, and **5a–c** carrying the same α-substituent (R^3^) was so insubstantial that it is groundless to speak about the essential effect of substituents on the nitrogen atom (R^1^, R^2^) on catalytic activity.

The high activity of ruthenium derivatives **4–5** in RCM reactions prompted us to investigate its behavior in metathesis polymerization reactions. As is known, this is one of the more often used processes in the polymer industry. For this purpose, the ring-opening metathesis polymerization (ROMP) of *trans*-dibutoxycarbonyl substituted bicyclo[2.2.1]hept-2-ene (5-*endo*-6-*exo*-di(*n*-butoxycarbonyl)norbornene, **DBNB**, **9**), a monomer which is liquid under normal conditions, was chosen (Figure 7 and Figure 6). The thermal effects of the polymerization of **9** to **10** are the simplest and most easily observable, which makes it possible to estimate the rate of the process.

A series of catalysts **4a**, **4d**, **5a**, and **5d** was tested for comparative study of the polymerization of **DBNB**. To study the ROMP, in each range, catalysts with the highest (**4d**, **5d**) and lowest (**4a**, **5a**) steric load were selected. A reaction was carried out under constant stirring (450 rpm) in a 30 mL glass vessel, with the temperature being controlled using two thermocouples (one in a water bath and another in a reaction vessel). The catalysts were added as solutions in DCM, and, for convenience of analysis, there was constant video recording of the temperature sensor readings.

Under the selected conditions, polymerization reaction started after about 3–12 min (Figure 6). The highest temperature of the exotherm was detected for sterically hindered complex **5d**, and the lowest one, **4a**, possessed the smallest substituents in all positions. From the performed experiments, it can be concluded that the steric load on the nitrogen atom plays a more significant role in the activation of the complex than the substituents in the α-position.

## 3. Materials and Methods

### 3.1. General Remarks

All reagents were purchased from commercial suppliers (Acros Organics, Morris Plains, NJ, USA and Merck KGaA, Darmstadt, Germany) and used without further purification. The metathesis reactions required solvents (CH_2_Cl_2_ and CHCl_3_) pre-dried over anhydrous P_2_O_5_ and an inert atmosphere (dry Ar). The thin layer chromatography was carried out on aluminum-backed silica precoated plates «Sorbfil» or «Alugram». The plates were visualized using a water solution of KMnO_4_ or UV (254 nm). After extraction, organic layers were dried over anhydrous MgSO_4_. IR spectra were obtained in KBr pellets or in thin films using an Infralum FT-801 or Nicolet 6700 IR-Fourier spectrometers. The NMR spectra were run in deuterated solvents (>99.5 atom % D) on a Jeol JNM-ECA 600 (600.1 MHz for ^1^H and 150.9 MHz for ^13^C) spectrometer for 2–5% solutions in CDCl_3_ at 22–23 °C using residual solvent signals (7.26/77.0 ppm for ^1^H/^13^C in CDCl_3_) or TMS as an internal standard.

### 3.2. Experimental Procedures

The isoquinoline derivatives were synthesized using the Bischler–Napieralski reaction according to procedures described earlier [39].

The detailed methods for the preparation of new compounds obtained in this work is given in the ESI section.

### 3.3. Single-Crystal XRD Analysis

The crystal structure of all synthesized substances was determined with X-ray structural analysis using an automatic four-circle area-detector diffractometer, the Bruker KAPPA APEX II with MoKα radiation. The crystal data, data collection, and structure refinement details are summarized in Appendix A (see the ESI). All other crystallographic parameters of the structures are indicated in Appendix A (see the ESI). The atomic coordinates were deposited at the Cambridge Crystallographic Data Centre (CCDC) [44]. The CCDC numbers are 2232047–2232052 for **4a**, **4b**, **4c**, **5a**, **5b**, and **5d**, respectively. The Supplementary crystallographic data can be obtained free of charge from the Cambridge Crystallographic Data Centre via www.ccdc.cam.ac.uk/data_request/cif (accessed on 21 December 2022).

## 4. Conclusions

In this work, we reported on the synthesis of a series of second-generation HG-type complexes containing a nitrogen–ruthenium coordinate bond in the six-membered chelate ring, in which an alkyl group (Me, Et, *^i^*Pr) is attached to the α-position of the benzylidene fragment. The majority of the aforementioned metal complexes can be synthesized easily and in excellent yields based on the reaction between available (2-vinyl)benzylamines and **Ind II**. It has been experimentally shown that the steric volume of the substituents at the nitrogen atom and in the α-position of the 2-vinylbenizamine ligands has a decisive influence on the possibility of assembly of the chelate ring. The most sterically loaded ligand suitable for chelate synthesis is N,N-diethyl-α-ethyl-2-vinylbenzylamine. An isopropyl group being introduced into the benzylamine ligand does not allow the chelate ring to close and makes the existence of the corresponding ruthenium complex impossible.

Activity of the target ruthenium chelates towards the simplest ROMP and RCM reactions was investigated in the concentration range of 0.1–0.01 mol%. It has been proven that the substituents attached to the nitrogen atom have the greatest influence on the rate of metathesis. The catalysts bearing an N-morpholine moiety manifested the highest catalytic activity. The methyl or ethyl groups in the α-position did not significantly affect the rate of the metathesis reactions. The storage stability and the catalytic properties of the ruthenium chelates turned out to be comparable to those of commercially available metathesis catalysts (**HG-II**).

Thus, the aforementioned range of ruthenium complexes completes our research [28,29], which is devoted to the synthesis and properties of six-membered HG-type complexes containing an N→Ru bond in the chelate ring.

## Data Availability

“MDPI Research Data Policies” at https://www.mdpi.com/ethics.

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
