# Peer review of "Influence of Substituents in a Six-Membered Chelate Ring of HG-Type Complexes Containing an N→Ru Bond on Their Stability and Catalytic Activity"

_molecules, 2023, doi:10.3390/molecules28031188_

Round 1
Reviewer 1 Report
General comment
In the manuscript named “Influence of substituents in a six-membered chelate ring of HG-type complexes containing an N→Ru bond on their stability and catalytic activity”, the Authors describe the synthesis of new second-generation Hoveyda-Grubbs-type catalysts containing a nitrogen-ruthenium coordinate bond. The catalytic properties of the complexes were investigated in ROMP and RCM reactions. Overall, this is a study that provides some insight into the possible methodologies to obtain this class of compounds as well as the catalytic applications. Publication in Molecules can be recommended after revisions are made/considered.
Specific comment
The authors may consider adding further detail to the characterization of the complexes using other techniques, such as NMR.
Author Response
- The authors may consider adding further detail to the characterization of the complexes using other techniques, such as NMR.
Response: We added the following sentence in the main part of the manuscript.
“The most well-defined signals of the key Ru=CH fragment are observed in the range of δ 18.70−19.00 ppm in 1H NMR and 211.1−213.6 ppm in 13C NMR spectra. Such downfield shifts are usual for chelates with a coordinating N→Ru bond [8,28,29,36] and are explained by the strong polarization of the Ru=C bond towards the metal atom. Otherwise, the obtained NMR spectra do not differ from those already described earlier for homologues [28,29]. It should be noted that signals of the mesityl fragments (Mes) in the both proton and carbon spectra turn out to be strongly broadened due to a slow rotation.”
Reviewer 2 Report
In the paper entitled "Influence of substituents in a six-membered chelate ring of HG-type complexes containing an N→Ru bond on their stability and catalytic activity", Vasilyev et al. report the synthesis some styrenic derivatives as well as some of the corresponding Ru-complexes.
These complexes were examined for their catalytics properties in the ring-closing metathesis (RCM).
The paper looks very sound and could be of interest for the readers of "Molecules" journal. some minor suggestions are below:
1- In table 1: sentences "acceptable for the cyclization" and "the cyclization is impossible" could be added as footnotes below the table.
2- The same remark for Table 2.
3- In the supplementary materials, the MS and FTIR spectra should be
provided.
4- In the supplementary materials, I think that no 31P NMR spectra were recorded
Author Response
We are eternally grateful to the reviewer for taking the trouble to read our work. We tried to take into account all the noted shortcomings. All corrections were highlighted in yellow in the main text of the paper.
- In table 1: sentences "acceptable for the cyclization" and "the cyclization is impossible" could be added as footnotes below the table.
Response: If possible, we would like to leave the table as it is. Transfer of the sentences to the bottom of the table will result in the appearance of empty cells. This will not look properly. We made some changes in Tables 1 and 2. In our opinion, the meaning of the sentences became clearer.
- The same remark for Table 2.
Response: Of course, if the reviewers and the editor insist, we will redo Tables 1 and 2.
- In the supplementary materials, the MS and FTIR spectra should be provided.
Response: Copies of MS and FTIR spectra for new compounds were added to the ESI.
- In the supplementary materials, I think that no 31P NMR spectra were recorded
Response: Thank you; we recorded 31P NMR spectra for the initial complex Ind II to check its purity. However, indeed, we did not enclose these spectra to the ESI (previously they were published many times). The mention of 31P NMR spectra was removed from the ESI.
Reviewer 3 Report
Report on the manuscript entitled “Influence of substituents in a six-membered chelate ring of HG-2 type complexes containing an N→Ru bond on their stability and catalytic activity” by Polyanskii and coworkers.
The authors present the synthesis and catalytic activity of a series of new ruthenium-based olefin metathesis catalysts. These catalysts recall known catalysts already published by the same authors in 2019 in Beilstein J. Org. Chem, but slight modifications of the structures in the alpha-N position have been introduced and the impact on the catalytic performances studied. In the end, it turns out that those structural modifications do not bring any improvement of the catalytic performances which are essentially linked to the substituents on the Nitrogen atom. However, the present manuscript is a nice piece of synthetic work that disserves publication. Nevertheless, a few issues must be considered before publication.
- The drawings of the ruthenium catalysts should be improved
o The chloro ligand should be drawn in a trans arrangement
o It is no longer in use to display the NHC and phosphino-ligand bonds to Ru with an arrow
o A bow should be included in the NHC ligand over the NCN atoms
- The tables are well presented excepted, in my opinion, in two places where results are a bit confusing
o Table 1 & 2 entry d ; possible or not?
On page 6, the second section between l163 and l 168 is confusing. I think one sentence should be moved to the previous section “ Probably the presence of 3-ethyl…
Same page about XRD analysis. The authors mentions intramolecular hydrogen bonds..see ESI
This sounds interesting but ESI does not help identifying the hydrogen bonds. Tables are provided but without ORTEP views of the complexes, they are useless. ORTEP views should be inserted.
The catalytic studies section is a bit disappointing. Contrary to their previous work in Beilstein J Org Chem, the authors have herein studied or presented the RCM of diallyltosylamide only along with a ROMP example. The RCM of diallyltosylaminde is actually one of the easiest RCM to conduct. It was essentially used in the early days of RCM. Nowadays, RCM studies should be conducted at least with DEDAM and also include a cross-metathesis example. Does the absence of such results in the present manuscript mean that the new catalysts are not or poorly active in such transformations? If yes, the results should be provided as limitations that could be of interest for readers.
ESI: NMR analysis should be carefully checked in particular 13C NMRs as carbon count is sometimes incorrect.
Author Response
We are eternally grateful to the reviewer for taking the trouble to read our work. We tried to take into account all the noted shortcomings. All corrections were highlighted in yellow in the main text of the paper.
The drawings of the ruthenium catalysts should be improved
- The chloro ligand should be drawn in a trans arrangement
Response: Scheme 5 and Table 2 were redrawn. Now the both chlorine atoms occupy the trans-position.
- It is no longer in use to display the NHC and phosphino-ligand bonds to Ru with an arrow. A bow should be included in the NHC ligand over the NCN atoms.
Response: We do not quite agree with the reviewer regarding this issue. Based on the materials of recent publications (10.1002/adsc.202101515, 10.1002/anie.202014929, 10.1002/anie.202201472, 10.1016/j.jorganchem.2022.122320 and others), it can be concluded that using of arrows to display an NHC-metal bond is acceptable. On the contrary, the use of a bow in the same cases is less common. In any case, as far as we know, the IUPAC rules do not prohibit the use of an arrow to indicate a coordination bond. Thus, we have kept the structure formulas unchangeable in this part.
- The tables are well presented excepted, in my opinion, in two places where results are a bit confusing. Table 1 & 2 entry d ; possible or not?
Response: We made some changes in Tables 1 and 2. In our opinion, the meaning of the sentences became clearer.
- On page 6, the second section between l163 and l 168 is confusing. I think one sentence should be moved to the previous section “Probably the presence of 3-ethyl… Same page about XRD analysis. The authors mentions intramolecular hydrogen bonds..see ESI. This sounds interesting but ESI does not help identifying the hydrogen bonds. Tables are provided but without ORTEP views of the complexes, they are useless. ORTEP views should be inserted.
Response: We have added Figures S5−S10 showing intramolecular hydrogen bonds in the ESI part. The sentence was rephrased more clearly as “Probably, the simultaneous presence of three ethyl groups in the benzylidene moiety of complex …”
- The catalytic studies section is a bit disappointing. Contrary to their previous work in Beilstein J Org Chem, the authors have herein studied or presented the RCM of diallyltosylamide only along with a ROMP example. The RCM of diallyltosylaminde is actually one of the easiest RCM to conduct. It was essentially used in the early days of RCM. Nowadays, RCM studies should be conducted at least with DEDAM and also include a cross-metathesis example. Does the absence of such results in the present manuscript mean that the new catalysts are not or poorly active in such transformations? If yes, the results should be provided as limitations that could be of interest for readers.
Response: The estimated reviewer is right. Catalytic properties are not the strongest side of the present work. The main goal of this paper was an assessment of the possibility of existence of ruthenium complexes with a high steric load near a nitrogen atom. We suppose that the catalytic behavior of the obtained catalysts will be close to the similar chelates obtained earlier in Beilstein J. Org. Chem. Therefore, we performed only kinetic studies for the simplest model reaction of RCM and compared the catalytic efficiency of the resulting complexes with its closest analogues. The RCM of N,N-diallyltosylamide was chosen because the reaction proceeds smoothly without formation of any by-products, which makes it possible to unambiguously evaluate the conversion.
In the previous works, despite using of a wide range of metathesis substrates, only conversion was determined, which does not describe entirely the kinetics of the process.
- ESI: NMR analysis should be carefully checked in particular 13C NMRs as carbon count is sometimes incorrect.
Response: Thank you, all NMR part in the ESI was rechecked and corrected.
Reviewer 4 Report
Reviewer’s Comments:
The manuscript “Influence of substituents in a six-membered chelate ring of HG-type complexes containing an N→Ru bond on their stability and catalytic activity” is very interesting work. An efficient approach to the synthesis of new olefin metathesis catalysts containing an N→Ru bond in a six-membered chelate ring was proposed. It was experimentally proved that with the increase of the steric volume of substituents around a nitrogen atom and in the α-position, the stability of ruthenium complexes dramatically decreases. The bulkiest iPr substituents attached to the nitrogen atom or to α-position do not allow the chelate cycle to close, making it impossible to obtain such type of sterically overloaded chelates. Catalytic activity of the complexes in RCM reactions compared with analogical α-unsubstituted previously obtained catalysts was also tested. However, the following issues should be carefully treated before publication.
1. In abstract, the author should add more scientific findings.
2. Keywords: the synthesized system is missing in the keywords. So, modify the keywords.
3. In the introduction part, the introduction part is not well organized and cited references should cite recently published articles such as 10.3390/molecules27196457, 10.3390/molecules27196564
4. Introduction part is not impressive and systematic. In the introduction part, the authors should elaborate the scientific issues in the catalytic activity research.
5. Synthesis of the initial 2-vinylbenzylamines 1, 2, and 3…, The author should provide reason about this statement “Based on the available starting compounds, two synthetic routes to the target 2-vinyl benzylamines (1-3) have been developed”.
6. The authors should explain regarding the recent literature why “Therefore, an alternative synthetic approach has been elaborated for preparation of α-ethyl substituted styrenes 3 (Scheme 3)”.
7. Synthesis of ruthenium complexes. The author should explain the latest literature “These complexes are poorly crystallized after cooling of the reaction mixtures and the resulting precipitates are soluble even in ice-cold pentane”.
8. The author should provide reason about this statement, “In all complexes, the chlorine atoms are coordinated to the ruthenium atom in the trans-position”.
9. Comparison of the present results with other similar findings in the literature should be discussed in more detail. This is necessary in order to place this work together with other work in the field and to give more credibility to the present results.
10. The conclusion part is very week. Improve by adding the results of your studies.
Author Response
Reviewer 3. Comments and Suggestions for Authors
We are eternally grateful to the reviewer for taking the trouble to read our work. We tried to take into account all the noted shortcomings. All corrections were highlighted in yellow in the main text of the paper.
- In abstract, the author should add more scientific findings.
Response: The abstract was rewritten and extended.
- Keywords: the synthesized system is missing in the keywords. So, modify the keywords.
Response: Keywords were added.
- In the introduction part, the introduction part is not well organized and cited references should cite recently published articles such as 10.3390/molecules27196457, 10.3390/molecules27196564
Response: The first mentioned paper was added as reference 12, the second one, Khan, S.; Iqbal, S.; Rahim, F.; Shah, M.; Hussain, R.; Alrbyawi, H.; Rehman, W.; Dera, A. A.; Rasheed, L.; Somaily, H. H.; Pashameah, R. A.; Alzahrani, E.; Farouk, A.-E. New Biologically Hybrid Pharmacophore Thiazolidinone-Based Indole Derivatives: Synthesis, In Vitro Αlpha-Amylase and Αlpha-Glucosidase Along with Molecular Docking Investigations. Molecules 2022, 27 (19), 6564. https://doi.org/10.3390/molecules27196564 , in our opinion, is not suitable for citation in this work.
- Introduction part is not impressive and systematic. In the introduction part, the authors should elaborate the scientific issues in the catalytic activity research.
Response: We have rewritten the introduction.
- Synthesis of the initial 2-vinylbenzylamines 1, 2, and 3…, The author should provide reason about this statement “Based on the available starting compounds, two synthetic routes to the target 2-vinyl benzylamines (1-3) have been developed”.
Response: Two close methods have been proposed for synthesis of styrenes 1−3. In the first case, needed benzylidene ligands were synthesized starting from available N-acylated phenethylamines or 3,4-dihydroisoquinoline. To obtain morpholine-substituted styrenes 1d and 3d (Scheme 4), a different approach based on the double alkylation of available 1,2,3,4-tetrahydroisoquinolines was used. Note that both methods can be used on an industrial scale. The mentioned paragraph was rephrased.
- The authors should explain regarding the recent literature why “Therefore, an alternative synthetic approach has been elaborated for preparation of α-ethyl substituted styrenes 3 (Scheme 3)”.
Response: This paragraph provides links to the most recent and relevant publications (40 and 41). This sequence of reactions has not previously been described, since the target styrenes 1-3 have a limited range of applications. All three approaches to the synthesis of styrenes 1−3 shown in Schemes 1, 3 and 4 complement each other. Depending on the substituents needed and the starting reagents available, a chemist may use the most convenient route. This paragraph was added to page 4.
- Synthesis of ruthenium complexes. The author should explain the latest literature “These complexes are poorly crystallized after cooling of the reaction mixtures and the resulting precipitates are soluble even in ice-cold pentane”.
Response: Unclear remark. In our opinion, the physical properties of new compounds should not be confirmed or refuted by literature data.
- The author should provide reason about this statement, “In all complexes, the chlorine atoms are coordinated to the ruthenium atom in the trans-position”.
Response: According to the results of XRD analysis, the angle between the chlorine atoms is approximately 180 degrees. All complexes have a similar spatial structure. We have decoded this statement in the main part of the paper and redrew structures of all complexes in the Table 2 and in the Scheme 5.
- Comparison of the present results with other similar findings in the literature should be discussed in more detail. This is necessary in order to place this work together with other work in the field and to give more credibility to the present results.
Response: We rewrote the introduction in concordance with the remarks of the referee.
- The conclusion part is very week. Improve by adding the results of your studies.
Response: The conclusion was improved.
Reviewer 5 Report
This article is devoted to the preparation of ruthenium complexes with various organic substituents. The article is promising and a large amount of experimental data make a good impression. There are some points that would like to be improved:
1. It is desirable to make the rationale for obtaining these ruthenium-containing substances more weighty.
2. Abstract needs to be expanded.
3. It is desirable to make more comparisons with literary sources. This will give more validity to some of the conclusions.
4. It is desirable to justify the choice and prospects of using ruthenium in synthesis and catalysis. At this point, it is desirable to cite the work: 10.3390/catal12111384.
5. It is desirable to restructure the conclusions.
In general, the article is interesting and can be accepted for publication after minor revision.
Author Response
We are eternally grateful to the reviewer for taking the trouble to read our work. We tried to take into account all the noted shortcomings. All corrections were highlighted in yellow in the main text of the paper.
- It is desirable to make the rationale for obtaining these ruthenium-containing substances more weighty.
Response: We rewrote the introduction in concordance with the remarks of the referee. The validity of this study is supported as well by references 28−38.
- Abstract needs to be expanded.
Response: The abstract was expanded.
- It is desirable to make more comparisons with literary sources. This will give more validity to some of the conclusions.
Response: The conclusion was improved.
- It is desirable to justify the choice and prospects of using ruthenium in synthesis and catalysis. At this point, it is desirable to cite the work: 10.3390/catal12111384.
Response: This reference was added in the introduction (see reference 14).
- It is desirable to restructure the conclusions.
Response: The conclusion was totally rewritten.